# Leucine Reduced Blood–Brain Barrier Disruption and Infarct Size in Early Cerebral Ischemia-Reperfusion

**DOI:** 10.3390/brainsci13101372

**Published:** 2023-09-26

**Authors:** Oak Z. Chi, Xia Liu, Jedrick Magsino, Harvey R. Weiss

**Affiliations:** 1Department of Anesthesiology and Perioperative Medicine, Rutgers Robert Wood Johnson Medical School, 125 Paterson Street, Suite 3100, New Brunswick, NJ 08901-1977, USA; liuxi@rwjms.rutgers.edu; 2Department of Biochemistry and Molecular Biology, Rutgers Robert Wood Johnson Medical School, 675 Hoes Lane West, Piscataway, NJ 08854-8021, USA; jam1001@scarletmail.rutgers.edu; 3Department of Neuroscience and Cell Biology, Rutgers Robert Wood Johnson Medical School, 675 Hoes Lane West, Piscataway, NJ 08854-8021, USA; hweiss@rwjms.rutgers.edu

**Keywords:** cerebral ischemia-reperfusion, neuroprotection, blood–brain barrier, leucine, Akt, ribosomal protein S6

## Abstract

A disruption of the blood–brain barrier (BBB) is a crucial pathophysiological change that can impact the outcome of a stroke. Ribosomal protein S6 (S6) and protein kinase B (Akt) play significant roles in early cerebral ischemia-reperfusion injury. Studies have suggested that branched-chain amino acids (BCAAs) may have neuroprotective properties for spinal cord or brain injuries. Therefore, we conducted research to investigate if leucine, one of the BCAAs, could offer neuroprotection and alter BBB disruption, along with its effects on the phosphorylation of S6 and Akt during the early phase of cerebral ischemia-reperfusion, specifically within the thrombolytic therapy time window. In rats, ten min after left middle cerebral artery occlusion (MCAO), 5 µL of 20 mM L-leucine or normal saline was injected into the left lateral ventricle. After two hours of reperfusion following one hour of MCAO, we determined the transfer coefficient (K_i_) of ^14^C-α-aminoisobutyric acid to assess the BBB disruption, infarct size, and phosphorylation of S6 and Akt. Ischemia-reperfusion increased the K_i_ (+143%, *p* < 0.001) and the intra-cerebroventricular injection of leucine lowered the K_i_ in the ischemic-reperfused cortex (−34%, *p* < 0.001). Leucine reduced the percentage of cortical infarct (−42%, *p* < 0.0001) out of the total cortical area. Ischemia-reperfusion alone significantly increased the phosphorylation of both S6 and Akt (*p* < 0.05). However, the administration of leucine had no further effect on the phosphorylation of S6 or Akt in the ischemic-reperfused cortex. This study suggests that an acute increase in leucine levels in the brain during early ischemia-reperfusion within a few hours of stroke may offer neuroprotection, possibly due to reduced BBB disruption being one of the major contributing factors. Leucine did not further increase the already elevated phosphorylation of S6 or Akt by ischemia-reperfusion under the current experimental conditions. Our data warrant further studies on the effects of leucine on neuronal survival and its mechanisms in the later stages of cerebral ischemia-reperfusion.

## 1. Introduction 

Following the initial brain damage caused by a stroke, even if the blood flow is restored within a few hours with or without thrombolytic therapy, the brain cells may still sustain further damage or perish due to ischemia-reperfusion injury. During this period of ischemia-reperfusion, the blood–brain barrier (BBB) may be disrupted by various factors, including glutamate, inflammatory factors, calcium, and free radicals. They may affect functionally, as well as anatomically, any component of the neurovascular unit or BBB that is composed of tight junctions, endothelium, astrocytes, pericytes, microglia, and neurons [1,2,3,4]. Minimizing BBB disruption, which is a major injury resulting from ischemia-reperfusion, may lead to a more favorable neurological outcome following a stroke [5,6,7,8].

Branched-chain amino acids (BCAAs), such as leucine, isoleucine, and valine, are involved in the synthesis of proteins, the modulation of neurotransmitters, signaling pathways, and the production of energy in the brain [9,10,11,12]. There have been reports that BCAAs can exert protective effects in ischemia-reperfusion damage in the heart and liver [13,14,15]. However, studies of the effects of BCAAs on the survival of neurons in early ischemia-reperfusion in the brain are lacking. 

In acute cerebral ischemia, the BCAAs in plasma and cerebrospinal fluid are decreased. Lower BCAA levels suggest a poor neurological outcome in stroke [16]. BCAAs have yielded beneficial effects on the axonal regeneration of injured retinal ganglion cells following optic nerve transection [17]. Leucine has promoted axonal outgrowth and regeneration in spinal cord injury [18]. Dietary BCAAs have ameliorated the cognitive impairment induced by brain injury [19]. These studies suggest that BCAAs might help in reducing cerebral ischemia-reperfusion injury.

Studies have shown that BCAAs can decrease inflammation and microcirculation failure in hepatic ischemia-reperfusion injury [13]. Based on these findings, we hypothesized that leucine, one of the BCAAs, may have the ability to reduce blood–brain barrier disruption during cerebral ischemia-reperfusion and increase neuronal survival. The timely administration of BCAAs provides protection against ischemia-reperfusion injury in the heart [14]. However, this BCAA-induced cardiac protection was prevented by the mechanistic target of rapamycin (mTOR) inhibitors or rapamycin. Interestingly, wortmannin, an Akt inhibitor, was not effective at reducing the cardioprotective effects induced by BCAAs [14]. Leucine is involved in the turnover of proteins and glutamate, as well as mTOR complex 1 (mTORC1) signaling [9,10,14,20,21]. In a previous study, we demonstrated the increased activation of ribosomal protein S6 (S6) and Akt during early cerebral ischemia-reperfusion [22,23,24]. Since various inhibitors or activators of the Akt-mTOR-S6 axis affected the degree of BBB disruption and neuronal survival within a few hours of cerebral ischemia-reperfusion [22,25,26,27,28], we conducted this study to examine whether an acute elevation of leucine levels in the brain can confer neuroprotection during the early phase of ischemia-reperfusion within a few hours of stroke. Additionally, we aimed to investigate the impact of leucine on BBB disruption, as well as the activities of S6 and Akt.

In this study, at two hours of reperfusion, after one hour of middle cerebral artery occlusion (MCAO), the transfer coefficient (K_i_) of ^14^C-α-aminoisobutyric acid (^14^C-AIB) and volume of ^3^H dextran (70,000 Da) were determined to assess the functional degree of BBB disruption. The size of the infarct was determined using tetrazolium staining. The phosphorylation of S6 (Ser240/244) and phosphorylation of Akt (Ser473) were determined to assess the activities of S6 and Akt. Our findings indicated that an intracerebroventricular injection of leucine following MCAO led to a reduction in the size of the infarct, accompanied by a decrease in BBB disruption. Furthermore, the process of ischemia-reperfusion itself resulted in increased levels of phosphorylation of both pS6 and pAkt. However, the injection of leucine did not further impact the phosphorylation of S6 or Akt.

## 2. Materials and Methods

We followed the US Public Health Service Guidelines and the Guide for the Care of Laboratory Animals (DHHS Publication No. 85-23, revised 1996). Our Institutional Animal Care and Use Committee approved this study (Approval number: 202200045).

### 2.1. Animals

Male Fischer 344 rats (Charles River Laboratories, Wilmington, MA, U.S.A.) were used in this experiment. The rodents were subjected to a 12 h light and 12 h dark cycle, and they had free access to standard rodent chow and water *ad libitum*. They weighed 220–250 g. Two groups were formed randomly: (1) Control Group (MCAO/reperfusion + vehicle, n = 17, vehicle was normal saline), and (2) Leucine Group (MCAO/reperfusion + leucine, n = 17). As we described previously [22,26,27], to anesthetize them, 2% isoflurane with a mixture of oxygen and air was used, and they were ventilated through tracheal tubes. A femoral artery and afemoral vein were catheterized. To monitor the heart rate and blood pressure, the femoral artery catheter was connected to a Statham P23Db pressure transducer and an Iworx data acquisition system throughout the experimental period (one hour of MCAO and two hours of reperfusion). Arterial blood samples from the femoral artery were obtained to determine the hemoglobin, blood gases, and pH using a Radiometer blood gas analyzer (ABL80). Whenever necessary throughout the experimental period, arterial blood gas data were used to adjust the mechanical ventilation to maintain PaCO_2_ at around 40 mmHg. During the determination of the BBB permeability, blood samples were collected from the femoral artery. Radioactive tracers or normal saline were administered through the femoral venous catheter. Maintenance fluid (normal saline) was given at the rate of 4 mL/kg/hour. The blood loss caused by the blood gas analysis and sampling was replenished with a normal saline solution at three times the volume of the blood loss. To monitor the body temperature, a thermistor probe per rectum, which was servo-controlled, was used. The body temperature was maintained at 37.1 °C ± 0.4 with a heating lamp. A thermocouple probe (Omega Engineering, Inc., Stamford, CT, USA) was used to monitor the temporalis muscle for a representative pericranial temperature, which was maintained at 36.7 °C ± 0.5. The concentration of isoflurane was maintained at 1.4% after MCAO was performed and the placement of a burr hole. The timeline for the procedures is shown in Figure 1. Vital signs and blood gases at two hours of reperfusion, after one hour of MCAO, and just before the BBB permeability measurements are shown in Table 1.

### 2.2. Transient Middle Cerebral Artery Occlusion (MCAO)

We performed the MCAO as we described previously [22,26,27], which was modified from methods that have been described elsewhere [29]. To expose the left common carotid artery, a midline ventral cervical incision was made. We used a 4.0 monofilament strand, the end of which was coated with silicone (Doccol MCAO Sutures, Doccol Corporation, Sharon, MA, USA.). It was then threaded through the left external carotid artery and pushed in by about 1.7 mm into the left internal carotid artery. When resistance was encountered, it was kept in place to induce MCAO. After 60 min of MCAO, the filament was withdrawn to the external carotid artery beyond the bifurcation, then the external carotid artery was ligated, and the filament was removed to allow reperfusion. After 120 min of reperfusion, we measured the permeability of the BBB and infarct size, and Western blots were obtained.

### 2.3. Intracerebroventricular Injection

As we described previously [27], a stereotaxic frame (Kopf Instruments, Tujunga, CA, USA) was used to place the rats in an operative position. At 0.8 mm posterior and 1.5 mm lateral to the bregma, a cranial burr hole (1 mm) was drilled. A 26-gauge needle was inserted perpendicularly through the burr hole and advanced 3.6 mm into the left lateral ventricle, as described previously [27,30]. For the Leucine Group, 10 min after the MCAO, the left intraventricular administration of a total of 5 µL (2.5 µL/min) of 20 mM of L-leucine (Thermo Fisher Scientific, Waltham, MA, USA) was performed. This dose was similar to what was used in a previous study by Laeger et al. [31], where the food intake in rats was suppressed for 24 h, and the same as that of another study by Dufour et al., where the distribution of leucine was reported in the ipsilateral cortex for two hours [10]. The same volume of the vehicle (normal saline) was injected into the left lateral ventricle in the Control Group.

### 2.4. Permeability of Blood–Brain Barrier 

In eight rats in each group, as we described previously, the BBB permeability was determined [22,26,27]. At two hours of reperfusion, after one hour of MCAO, 20 μCi of ^14^C-α-aminoisobutyric acid (^14^C-AIB, molecular weight 104 Da, Amersham, Arlington Heights, IL, U.S.A.) was injected rapidly and intravenously through the femoral venous catheter, and was then flushed with 0.5 mL of normal saline. Afterwards, during the next two minutes, a blood sample of about 40 µL was collected in the capillary tube at 20 s intervals from the femoral artery catheter, then every minute for the next eight min period. At five min, after the ^14^C-AIB injection, 20 μCi of ^3^H-dextran (molecular weight 70,000 Da, Amersham, Arlington Heights, IL, U.S.A.) was administered intravenously through the femoral venous catheter and 0.5 mL of normal saline flush was implemented. The rats were decapitated after collecting the last ten min arterial blood sample. The following brain regions were dissected: the ischemic-reperfused cortex (IR-C), contralateral cortex (CC), ipsilateral hippocampus (IH), contralateral hippocampus (CH), cerebellum (CBLL), and pons. Soluene^TM^ (Packard, Downers Grove, IL, USA) was used to solubilize the brain samples. The plasma was separated from the arterial blood samples via centrifugation. The plasma and brain samples were counted in a liquid scintillation counter, which was equipped for dual-label counting. Using carbon tetrachloride, quench curves were prepared, and all the samples were corrected for quenching automatically. Assuming a unidirectional transfer of ^14^C-AIB over a ten min period of the experiment, the *K_i_* for ^14^C-AIB was determined using the equation described previously [26,27,28,32]:$$K_{i}=\frac{Am-(Vp\times CT)}{\int_{\;o}^{\;T}Cp(t)dt}$$
where *Am* is the amount of ^14^C-AIB radioactivity in the tissue per gram and *Vp* is the volume of plasma retained in the tissue. It was determined from the ^3^H-dextran data and the following equation: *Vp* = *A*’*m*/*C*’*p*, where *A*’*m* is the amount of ^3^H-dextran radioactivity in the tissue per gram and *C*’*p* is the concentration of ^3^H-dextran in the plasma at the time of decapitation. *Cp*(*t*) is the arterial concentration of ^14^C-AIB over time *t* and *CT* is the arterial plasma concentration of ^14^C-AIB at the time of decapitation. In the equation used to determine *K_i_*, *Vp* × *CT* is a correction term that accounts for the label ^14^C retained in the vascular compartment of the tissue, *Am*.

### 2.5. Size of Infarction

As we described previously [22,27], in six rats from each group, coronal sections of the brain using a straight-edge razor blade were obtained, yielding 3–4 slices of approximately a 2–3 mm thickness [22,26,27]. The brain slices were incubated for 30 min in a 0.05% solution of 2,3,5-triphenyltetrazolium chloride (TTC) (Sigma-Aldrich, St. Louise, MO, USA) in PBS at 37 °C [33]. We used this low dose of TTC to minimize false positive or false negative results. A weighing dish was prepared by filling it with PBS. After three washes of a one min wash in PBS, each slice was placed on the weighing dish, which was placed on a dissection microscope, then covered with a clear slide. The cortical region of each slice was traced with a 0.3 mm marker. Cross-hatching was used to mark any infarcted areas over any areas not well-marked with tetrazolium stain. Using ImageJ v. 1.52 (National Institute of Health, Bethesda, MD, USA) software, the percentage of the infarcted cortical area relative to the total cortical area was determined after the slide of each brain slice was scanned, and the percentage of the cortical infarct in each brain slice was averaged to represent the mean percentage of the cortical infarct of that rat. 

### 2.6. Western Blot

For the Western blot, three rats were used from each of the Control and Leucine Groups. The brain tissues of the IR-C and CC were lysed in a radioimmunoprecipitation assay buffer (RIPA buffer), as described previously [22]. The tissue lysate was centrifuged for 30 min at 4 °C using 15,000× *g*. The protein levels were measured and sample concentrations were normalized to 1–5 mg/mL using a Bradford Assay. Total extracts of 20–30 µg were loaded in each lane. The proteins were resolved by SDS-PAGE. To detect the phospho- and total proteins, primary antibody (1:1000) incubation was performed at 4 °C overnight. Secondary antibody (1:5000) incubation was performed at room temperature for one hour. Antibodies to S6, pS6 (Ser240/244), Akt, and pAkt (Ser473) were purchased from Cell Signaling (Danvers, MA, U.S.A.).

### 2.7. Statistical Analysis 

Using two-way ANOVA with a general linear model from the SAS Institute (Cary, NC, USA), the K_i_, the volume of dextran distribution, and the vital signs were compared between the two groups and among the various brain regions that were examined. The Tukey test was used for multiple comparisons. The size of the cortical infarct was compared using an unpaired Student *t*-test. The protein levels were compared using two-way ANOVA followed by a Bonferroni post hoc test. All the data were expressed as means ± standard deviation. The level of significance was set as *p* < 0.05. 

## 3. Results

### 3.1. Vital Signs and Blood Gas Values

Table 1 shows the vital signs and blood gas values of the Control and Leucine Groups before determining the BBB permeability at two hours of reperfusion after one hour of MCAO. The values for the mean blood pressure and heart rate in the Control Group were in the physiological ranges for rats [34]. There were no significant differences between the two experimental groups, except that the heart rate was lower in the Leucine Group. The rats used for the Western blot and for measuring the infarct size had no statistically significant different values in terms of the vital signs and blood gases from those of the corresponding groups that were used in the measurements of the BBB permeability.

### 3.2. Transfer Coefficient (K_i_)

In the Control Group (Figure 2A), the K_i_ of the IR-C was significantly higher (+143%, *p* < 0.001) than that of the CC. The K_i_ of IH and CH were significantly lower than the IR-C (*p* < 0.001). The K_i_ of the CC, the IH, and the CH were significantly lower than the pons (*p* < 0.01) in the Control Group (Figure 2A). In the Leucine Group, the intraventricular injection of leucine significantly reduced the K_i_ in the IR-C (−34%, *p* < 0.001) when compared to the Control Group. The K_i_ of the IR-C was significantly higher than the K_i_ of the CC (+70%, *p* < 0.01). The K_i_ of the IH, the CH, and the cerebellum were significantly lower than the K_i_ of the IR-C in the Leucine Group (*p* < 0.01) (Figure 2A). 

### 3.3. Volume of Dextran Distribution 

There were no statistically significant differences in the volumes of dextran distribution between the Control and Leucine Groups in any of the brain regions that were studied, nor among the brain regions in each group (Figure 2B). 

### 3.4. Size of Infarction

The percentage of the infarcted cortical area in relation to the total cortical area at two hours of reperfusion, after one hour of MCAO, is shown in a box plot for each of the experimental groups (Figure 3A). The intraventricular injection of leucine significantly reduced the percentage of the cortical infarct out of the total cortical area (−42%, *p* < 0.0001). Figure 3B shows a representative brain section from a similar brain region of each of the Control Group (MCAO/reperfusion + vehicle) and the Leucine Group (MCAO/reperfusion + leucine). We used a low concentration (0.05%) of 2,3,5-triphenyl tetrazolium chloride to minimize false positive or false negative staining [33]. The tissue samples were photographed at two hours after reperfusion without any fixatives. The area of the infarcted cortex appeared pink, not white, which suggested that infarction was still going on (Figure 3B). 

### 3.5. Western Blot

The effects of the intraventricular leucine on the phosphorylation of S6 at Ser240/244, as well as the phosphorylation of Akt at Ser473, were examined. In the Control Group, the phosphorylation of both S6 and Akt was significantly elevated in the IR-C when compared to the CC (*p* < 0.005) at two hours of reperfusion, after one hour of MCAO. The leucine treatment did not significantly affect the phosphorylation of S6 or Akt either in the CC or the IR-C when compared to the Control Group (Figure 4). It is notable that there was a trend towards increased Akt and S6 protein levels upon leucine treatment, although this increase did not reach statistical significance.

## 4. Discussion


*Leucine decreased the infarct size in early cerebral ischemia-reperfusion*


This is the first study which showed that an acute increase in leucine in the brain could be neuroprotective in early cerebral ischemia-reperfusion that is within the time window of thrombolysis therapy, mostly 4.5–6 h after the onset of stroke [35]. Our findings align with earlier research that hinted at the potential neuroprotective benefits of leucine in the cases of brain, spinal cord, and optic nerve injuries [17,18,19]. To investigate the neuroprotective mechanisms of leucine, we assessed its impact on the extent of BBB disruption during early cerebral ischemia-reperfusion. To find the molecular mechanisms involved, we examined the phosphorylation of S6 and Akt.


*Leucine decreased the BBB disruption in early cerebral ischemia-reperfusion*


Our findings indicated that the primary mechanism of the neuroprotection provided by leucine was through its ability to decrease BBB disruption. This is consistent with a previous report by Kitagawa et al. [13], which demonstrated that leucine can reduce inflammation and microcirculatory failure in hepatic ischemia-reperfusion injury.

When compared to the Control Group, leucine decreased the Ki of 14C-AIB only in the ischemic-reperfused cortex. This difference was not observed in the other brain regions. Our data suggested that the intracerebroventricular injection of leucine could have mainly diffused to the ipsilateral cortex, as another previous study showed [10]. The volume of dextran distribution was the sum of the dextran remaining in the plasma and the dextran that crossed the BBB into the brain tissue. There was no difference between the two groups in the volume of dextran distribution in our study. This suggested that leucine induced a decrease in the BBB permeability, predominantly to small molecules such as α-aminoisobutyric acid (molecular weight 104 Da), without significantly affecting larger molecules such as dextran (molecular weight 70,000 Da). Even with this degree of decrease in BBB disruption, excitatory amino acid neurotransmitters, ions, agents, and toxins could have been limited to pass the BBB and contributed to protecting the neurons or brain cells from early ischemia-reperfusion injury. The effects of leucine on glutamate, inflammation, and mitochondrial protection could also have helped in reducing the BBB disruption [9,10,13,14,15]. In the cerebral ischemia-reperfusion in this study, the leucine did not significantly affect the S6 phosphorylation. Thus, whether the mTORC1-S6 signaling axis could mediate the involvement of the vascular endothelial growth factor (VEGF), hypoxia-inducible factor 1α (HIF-1α), and matrix metalloproteinases (MMPs), which play roles in increasing BBB disruption, needs further investigation [36,37,38,39].


*Leucine did not further increase the phosphorylation of S6 and Akt in early cerebral ischemia-reperfusion*


In these experimental conditions, at two hours of reperfusion, after one hour of MCAO, the phosphorylation of both S6 and Akt, which are the mTOR signaling effectors, was increased in the ischemic-reperfused cortex, as previously reported [22,24]. Given this robust increase in the S6 and Akt phosphorylation, it is possible that mTOR signaling was maximal after ischemia-reperfusion and was not increased further by the leucine injection. In contrast, other investigators have reported that the phosphorylation of mTOR and Akt significantly decreases after 24 h of reperfusion, which differs from the findings of this study [40]. This suggests that mTOR signaling may be affected by the duration of the reperfusion. Since there was a trend towards increased Akt and S6 protein levels upon the leucine treatment, it is also possible that leucine could stabilize or enhance the expressions of these proteins. The protein levels of Akt are regulated by different mechanisms and are sensitive to nutrient levels [41,42,43]. We administered leucine at the same dose used by Dufour et al. [10], where the distribution of leucine in the ipsilateral cortex was reported for two hours. The dose and route we used were similar to those in the study by Laeger et al. [31], which resulted in a 24 h suppression of food intake in rats.

A bolus intravenous injection of radioactive leucine caused a sharp increase in its concentration in the brain. However, the leucine level decreased rapidly to around 30% in the next 10 min and to less than 10% at 35 min after the injection [44]. In the brain, in vivo, the leucine concentration can be reduced in various ways, including incorporation into proteins, transfer via the sodium-dependent neutral amino acid transporter, BCAA transaminase, and branched-chain keto acid dehydrogenase [44,45]. Obviously, it is hard to maintain a constant leucine level in the brain through external administration. Additionally, the administration of an oral dose of 500 mg/kg of leucine did not increase pS6 (Ser 240/244) in the brain one hour later [46], but increased pS6 in the heart, liver, and muscles. Hence, it is possible that further increasing the leucine dosage would not enhance the S6 phosphorylation under the current experimental conditions. 

Since mTORC1/2 could have tissue-specific targets depending on the conditions and stimuli, the effects of leucine on additional targets of mTOR signaling, other than Akt and S6, warrant further investigation [47].


*Other possible mechanism of neuroprotection*


In the early stages of cerebral ischemia-reperfusion, or within a few hours after a stroke like in our experimental timeframe, excitotoxicity is one of the reasons for major neuronal damage [2,3,4]. We speculate that modulating the activity of glutamate by leucine could have played a significant role in neuroprotection in this study [9,10]. The administration of BCAAs has been shown to reduce inflammation and microcirculatory failure in ischemia-reperfusion injury of the liver [13], which may have contributed to the observed decrease in the blood–brain barrier disruption and infarct size in our study of cerebral ischemia-reperfusion. There is a report that the Ca^++^-induced swelling of the heart mitochondria in mice was abolished by BCAAs. The opening of the mitochondrial permeability transition pore (mPTP) was decreased by BCAA treatment, resulting in the prevention of mitochondria-mediated cell death [14]. In our study, leucine could have helped to maintain the mitochondrial integrity during early cerebral ischemia-reperfusion. Leucine induces cardioprotection in vitro by promoting mitochondrial function not only via mTOR, but also via Opa-1 signaling [15]. Thus, leucine may affect other intracellular signaling pathways to affect the outcome of stroke. In contrast, an excessive intake of BCAA might establish neurotoxic conditions such as excitotoxicity and could facilitate neurodegeneration [48]. A chronic accumulation of BCAA could exacerbate myocardial ischemia-reperfusion vulnerability [49]. Further studies are needed to investigate the mechanisms of neuroprotection afforded by leucine in early ischemia-reperfusion in the brain.


*Limitations of this study*


There are several limitations to our study. Firstly, all the experimental parameters were measured at two hours of reperfusion after one hour of MCAO. It is possible that obtaining parameters during the later stages of reperfusion could yield different results. Additionally, the infarcted area (Figure 3B) appeared pink instead of white, suggesting ongoing infarction. Other staining methods capable of revealing both cell necrosis and apoptosis, as well as the quantification of various biomarkers related to apoptosis and inflammation, could have provided valuable insights to support our data. No animals died following the occlusion of the middle cerebral artery, and all of them were included in the data analysis. This can be attributed to the mechanical ventilation, fluid replacement, and monitoring of vital signs provided during the experimental period. 

## 5. Conclusions

Our study suggested that an acute elevation of leucine in the brain could be neuroprotective during early cerebral ischemia-reperfusion, that is, within a few hours of stroke. The reduced disruption of the BBB by leucine may be an important factor that contributed to its neuroprotective effects in our study. Leucine did not further increase the already elevated phosphorylation of S6 or Akt by ischemia-reperfusion in the current experimental conditions. Our data warrant further studies on the effects of leucine on neuronal survival and their mechanisms in the later stages of cerebral ischemia-reperfusion.

## Figures and Tables

**Figure 1 brainsci-13-01372-f001:**
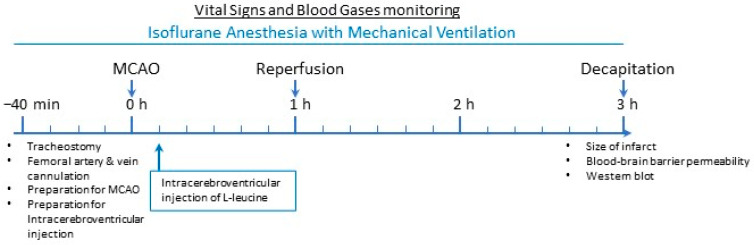
Timeline for the experimental procedures.

**Figure 2 brainsci-13-01372-f002:**
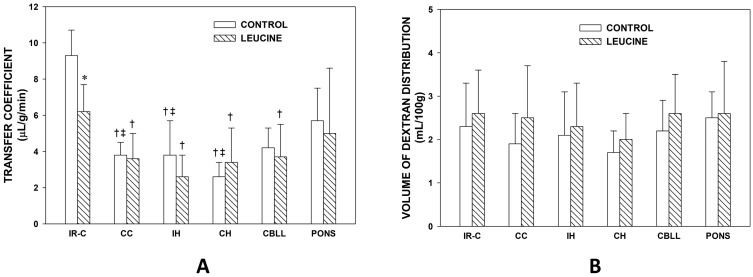
(**A**). Transfer coefficient (K_i_) of ^14^C-AIB in various brain regions of the MCAO/reperfusion + vehicle (Control Group) and MCAO/reperfusion + leucine (Leucine Group) after one hour of middle cerebral artery occlusion and two hours of reperfusion. n = 8 in each group. IR-C: Ischemic-reperfused cortex. CC: Contralateral cortex. IH: Ipsilateral hippocampus. CH: Contralateral hippocampus. CBLL: Cerebellum. *: *p* < 0.001 vs. Control Group. †: *p* < 0.01 vs. IR-C in the same group. ‡: *p* < 0.01 vs. pons in the same group. Values are means ± SD. (**B**). Volume of dextran distribution in various brain regions of the MCAO/reperfusion + vehicle (Control Group) and MCAO/reperfusion + leucine (Leucine Group) after one hour of middle cerebral artery occlusion and two hours of reperfusion. n = 8 in each group. CC: Contralateral cortex. IH: Ipsilateral hippocampus. CH: Contralateral hippocampus. CBLL: Cerebellum. Values are means ± SD.

**Figure 3 brainsci-13-01372-f003:**
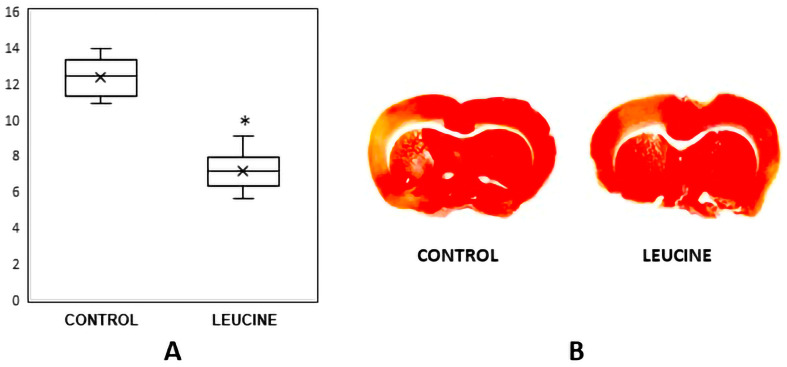
(**A**). Intracerebroventricular injection of leucine significantly decreased the percentage of the cortical infarcted area out of the total cortical area when compared to the control rats (n = 6, *p* < 0.0001). Each boxplot consists of 25th percentile, median, 75th percentile, and whiskers with the minimum and maximum. Mean is shown with the symbol “×”. *: *p* < 0.0001 vs. the Control Group. (**B**). A representative brain section from the similar brain region of each of the Control Group (MCAO/reperfusion + Vehicle) and the Leucine group (MCAO/reperfusion + leucine). The tissue samples were photographed at two hours after reperfusion without any fixatives. The area of the infarcted cortex appeared pink, not white, which suggests infarction is still going on.

**Figure 4 brainsci-13-01372-f004:**
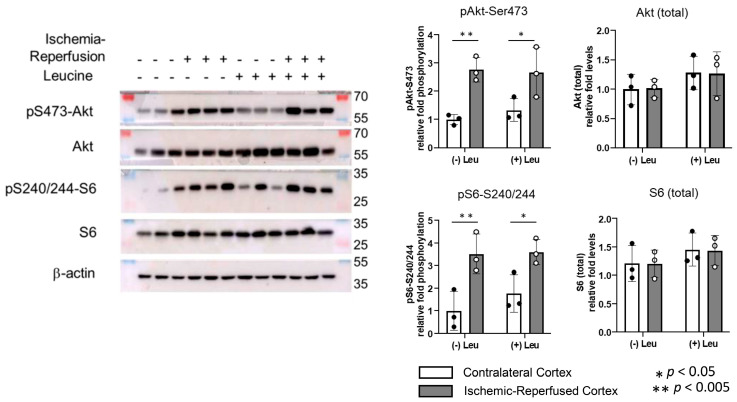
Representative Western blots of phosphorylation of S6 and phosphorylation of Akt, and their quantification after one hour of middle cerebral artery occlusion and two hours of reperfusion. Phosphorylation of Akt at Ser473 and phosphorylation of S6 at Ser240-244 were increased by ischemia-reperfusion. The intraventricular injection of leucine resulted in no significant changes in phosphorylation of S6 or Akt when compared with the Control Group in the ischemic-reperfused cortex (IR-C). n = 3. −Leu: without leucine treatment. +Leu: with leucine treatment. Values are means ± SD.

**Table 1 brainsci-13-01372-t001:** Hemodynamic and blood gas parameters for the control and the leucine-treated group at two hours of reperfusion after one hour of MCAO.

Group	Control Group (MCAO/Reperfusion + Vehicle)	Leucine Group (MCAO/Reperfusion + Leucine)
Mean blood pressure (mmHg)	84 ± 10	93 ± 19
Heart rate (beats/min)	350 ± 21	298 ± 53 *
Arterial PO_2_ (mmHg)	105 ± 25	122 ± 15
Arterial PCO_2_ (mmHg)	34 ± 8	35 ± 5
pH	7.35 ± 0.05	7.36 ± 0.09
Hemoglobin (g/100 mL)	9.6 ± 1.0	10.2 ± 1.3

MCAO: Middle cerebral artery occlusion. *: *p* < 0.05 vs. the Control Group. Values are mean ± SD.

## Data Availability

The raw data supporting the conclusions of this article will be made available upon request.

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
