# Peer review of "Leucine Reduced Blood–Brain Barrier Disruption and Infarct Size in Early Cerebral Ischemia-Reperfusion"

_brainsci, 2023, doi:10.3390/brainsci13101372_

Round 1

Reviewer 1 Report

The authors of the study titled “Leucine reduced blood-brain barrier disruption and infarct size in early cerebral ischemia-reperfusion“ aimed to determine if leucine, one of the BCAAs, could offer neuroprotection and alter BBB disruption, along with its effects on phosphorylation of S6 and Akt during the early phase of cerebral ischemia-reperfusion, specifically within the thrombolytic therapy time window. They observed that an acute increase in leucine levels in the brain during early ischemia-reperfusion within a few hours of stroke may offer neuroprotection, possibly due to reduced BBB disruption.

A.    The proficiency in the English language in this study is fine, minor errors detected.

B.     The title effectively captures the scope of the study.

C.     Abstract: The abstract provides a clear and understandable summary of the study, providing sufficient information to grasp its main objectives.

A.    Introduction: The introduction is adequately written, however, there are some issues.

·         “After initial brain damage due to stroke, although blood flow may be restored within a few hours with or without thrombolytic therapy, brain cells could continue to be damaged or die due to ischemia-reperfusion injury.” – This sentence is hard to follow.

·         “During this period of ischemia-reperfusion, the blood-brain barrier (BBB) may be disrupted by various factors, including glutamate, inflammatory factors, calcium, and free radicals”- This sentence appears incomplete, you mentioned glutamate, inflammatory factors, calcium, and free radicals…what are their mode/mechanisms of action?

·         “Minimizing blood-brain barrier (BBB) disruption, which is a major injury resulting from ischemia-reperfusion, may lead to a more favorable neurological outcome following a stroke [1–4].”- There is again BBB abbreviation.

·         “BCAAs enhanced survival of retinal ganglion cell and regeneration of axon after optic nerve damage [13]” – This sentence lacks clarity and coherence.

B.     The Materials & Methods: This section is thoroughly and well described. However, there are some issues that should be addressed.

·         “Our Institutional Animal Care and Use Committee approved this study.” – Herein should be stated a number of approval.

·         2.1. section- There is a dot before Animals

·         “Ad libitum” should be written in italics

·         In 2.1 should be stated what was used as a vehicle

·         You should choose if you are going to write USA after you mention states of USA. Whatever you prefer, just stick to it. (Kopf Instruments, Tujunga, CA; Doccol MCAO Sutures, Doccol Corporation, Sharon, MA, USA; Omega Engineering, Inc., Stamford, CT

·         Three rats for Western blot are not enough for statistics.

C.     Results: The results are effectively presented and comprehensively addressed in the manuscript.

·         3.1 There is a dot

D.    Discussion:

·         “Our data suggest that acute increase of leucine in the brain could be neuroprotective during early cerebral ischemia-reperfusion, that is, at two hours of reperfusion after one hour of MCAO which is within the time window of thrombolysis therapy that is mostly 4.5 - 6 hours after onset of stroke [31]. Our findings indicate that the primary mechanism of neuroprotection provided by leucine is through its ability to decrease BBB disruption.”- This paragraph could be merged with the previous one since some parts are repeating.

·         “The dose and route we used are similar to the study by Laeger et al. [27], which resulted in a 24-hour suppression of food intake in rats”- How is this related to your study?

·         “This suggests that mTOR signaling may be affected by the duration of reperfusion”. “Therefore, it is important to consider the timing of mTOR signaling analysis when interpreting the drug's impact on this pathway.” “It would be worth examining the effects of leucine on other mTOR targets and effectors, and at different time points of stroke. ”- All these sentences look alike.

·         VEGF, HIFα and MMPs -These are not defined previously in the text.

·         A substantial portion of the Discussion section relies heavily on the Western blots analysis of two specific proteins (their total and phosphorylated forms). Furthermore, the dataset consists of only three animals per experimental group. This sample size may be insufficient to draw comprehensive and definitive conclusions.

Overall, the paper “Leucine reduced blood-brain barrier disruption and infarct size in early cerebral ischemia-reperfusion “exhibits certain deficiencies in its writing quality, notably characterized by repetitive sentence structures and an over-reliance on a small sample size of three animals in a substantial portion of the Discussion section. Additionally, some sentences are quite confusing and difficult to understand. Although the work demonstrates potential, its realization hinges upon thorough refinement and reorganization. Consequently, I propose a major revision to enhance the manuscript's overall quality.

Minor editing required.

Reviewer 2 Report

Dear author,

Thanks for the submission. This manuscript describes the effect of the acute increase of leucine on reducing the percentage of cortical infarcts and preventing the BBB breakdown in the MCAO model in rats. However, there are some issues that need to be further addressed.

1:  In 2.1 Animal, Male Fischer 344 rats were used in this experiment, it should be 34, instead of 344, please make corrections accordingly.

2: In 2.2 Transient middle cerebral artery occlusion (MCAO), remove the extra hyphen in the last paragraph.

3: Please add more details in 2.4 Permeability of blood-brain barrier about how much blood sample was collected at each time point. And what did you use for analysis, whole blood, plasma, or serum? Please revise accordingly.

4: In 2.5 Size of infarction part, how to make sure the coronal sections are on the same plane of each animal? Whether Brain Matrix and Brain Slicers were used?  If so, please provide more details in 2.5 part.

5: Did the author evaluate the total infarction area throughout the whole brain or just from one single section? please address more details in the methods part. Also, in the results part, should show all the sectioned coronal sections throughout the brain.

6: Remove the extra hyphen in 2.7 Statistical analysis.

7: Remove the extra full stop mark in the 3.1 title part.

8: Need to uniform the font in the manuscript draft and in all the figures.

9: Need to replace the figures with higher resolution ones. 

Round 2

Reviewer 1 Report

The authors have answered to all of my questions, therefore, I propose accepting the paper.

Author Response

We greatly appreciate your review of our revision, and we want to extend our thanks for proposing the acceptance of the paper.

Reviewer 2 Report

Dear authors,

Thanks for the reply, however the quality of the figures are still pretty low, please consider replacing them with higher resolution ones. 

Author Response

We sincerely appreciate your review of our revision, and we extend our thanks for your valuable comments.

As explained in the Results and Figure Captions, the brain slices were stained with a low concentration of TTC, and the reperfusion time was short for producing pink infarcted areas. Additionally, in order to capture photographs, the brain slices were not fixed with any fixatives and were placed in a PBS solution to prevent glare. To measure the infarcted area, we traced the inadequately stained cortical region under a dissection microscope. Of note is that a photograph is not necessarily needed to calculate the percentage of cortical infarction. We apologize for not being able to provide high quality images. Nonetheless, we have made every effort to enhance the resolution of the presented photographs and we are resubmitting them. We hope these improved photographs meet your expectations and are acceptable.

Once again, thank you.